# Martensite Start Temperature Prediction through a Deep Learning Strategy Using Both Microstructure Images and Composition Data

**DOI:** 10.3390/ma16030932

**Published:** 2023-01-18

**Authors:** Zenan Yang, Yong Li, Xiaolu Wei, Xu Wang, Chenchong Wang

**Affiliations:** 1Science and Technology on Advanced High Temperature Structural Materials Laboratory, AECC Beijing Institute of Aeronautical Materials, Beijing 100095, China; 2State Key Laboratory of Rolling and Automation, Northeastern University, Shenyang 110819, China; 3School of Mechanical Engineering, Liaoning Petrochemical University, Fushun 113001, China

**Keywords:** martensite start temperature, microstructure, deep learning, data augmentation

## Abstract

In recent decades, various previous research has established empirical formulae or thermodynamic models for martensite start temperature (Ms) prediction. However, most of this research has mainly considered the effect of composition and ignored complex microstructural factors, such as morphology, that significantly affect Ms. The main limitation is that most microstructures cannot be digitized into numerical data. In order to solve this problem, a convolutional neural network model that can use both composition information and microstructure images as input was established for Ms prediction in a medium-Mn steel system in this research. Firstly, the database was established through experimenting. Then, the model was built and trained with the database. Finally, the performance of the model was systematically evaluated based on comparison with other, traditional AI models. It was proven that the new model provided in this research is more rational and accurate because it considers both composition and microstructural factors. In addition, because of the use of microstructure images for data augmentation, the deep learning had a low risk of overfitting. When the deep-learning strategy is used to deal with data that contains both numerical and image data types, obtaining the value matrix that contains interaction information of both numerical and image data through data preprocessing is probably a better approach than direct linking of the numerical data vector to the fully connected layer.

## 1. Introduction

Martensite start temperature (Ms) is a critical phase-transformation parameter for steel. This parameter could help to guide processing improvements in various advanced steels, such as third-generation advanced, high-strength steels [1,2,3,4]; ultrahigh-strength stainless steels [5,6,7]; and cryogenic steels [8,9,10]. Therefore, many studies in recent decades have focused on establishing and improving Ms prediction models, including their empirical formulae, thermodynamic equations, and machine learning strategies.

Linear empirical formulae are the most concise and commonly used methods for predicting Ms values. Various formulae for different alloy systems have been established, including the Andrews [11], Mahieu [12], and Trzaska [13] models. It should be noted that the internal mechanism of martensite transformation is much more complex than a linear relationship. Therefore, most linear empirical formulae have a relatively narrow application scope. In addition to the empirical models, thermodynamic models based on physical, constitutive equations have also been studied for decades [14,15,16,17]; for example, the Stormvinter [15] and Olson–Cohen [16,17] models. In particular, the Olson–Cohen model simplified the expression of frictional work using a series of semiempirical formulae, which partly evaded the controversial martensite transformation mechanism. The Olson–Cohen model has been successfully used for the design of various TRIP (transformation-induced plasticity) and high Co-Ni secondary hardening steels [18,19]. At the same time, because of the limitation of error accumulation in multicomponent alloy systems, most thermodynamic models can only be used for alloy systems with fewer than 10 elements.

Therefore, with continued development of advanced steels, more powerful methods, such as machine learning, which can consider high-dimensional factors [20,21,22,23], are needed for generic alloy design. In 2019, Rahaman et al. [23] reported a machine-learning model for Ms prediction. After feature engineering, 14 composition features were included in this Ms prediction model, which was much more generic than traditional empirical or thermodynamic models. After comparison with other models, the accuracy of this machine-learning model was proven to be better. In addition, to further improve the performance of machine-learning models, Lu et al. [22] combined thermodynamic knowledge and machine-learning methods. Thermodynamic features were used as input in multilayer feed-forward neural networks, and the modified machine learning models could consider 17 element features with high accuracy. However, all of the aforementioned models only focused on the effect of composition and ignored other important factors that affect Ms, such as grain size and austenite morphology [24,25].

To account for the effect of grain size on Ms in modeling, in previous studies, modifications were made to both the empirical formulae and the thermodynamic models. In 2007, Jimenez-Melero et al. [26,27] modified the Andrews model [11] through addition of a grain-size effect term to the linear equation, and the modified model was successfully used for Ms prediction in TRIP steels. In 2013, Lee et al. [28] established a more generic linear empirical formula for TRIP steels; it could account for the effect of grain size on Ms. In Lee’s model, a natural logarithm term for the grain-size effect was used instead of the index term used in Jimenez-Melero’s model. In addition, similar to the empirical formulae with only composition features, these modified linear empirical formulae also had a relatively narrow application scope. For thermodynamic models, Bohemen and Morsdorf [29] modified the description of frictional work in the Olson–Cohen model via addition of a Hall–Petch strengthening term to express the effect of grain size on the Ms. This model can be used for Ms prediction in a seven-element alloy system (Fe, C, Mn, Si, Cr, Ni, Mo). Although great efforts have been made towards modification of Ms prediction models, as mentioned above, most previous studies have focused only on modeling the effects of composition and grain size. At the same time, experiments have shown that Ms is related to various microstructural characteristics besides grain size, such as austenite morphology [30,31] and the surrounding phase of austenite [32,33]. In addition, these microstructural characteristics can only be expressed using image results, such as scanning electron microscopy images, and it is difficult to convert such images into numerical data that is suitable for modification of traditional empirical formulae or thermodynamic models. Therefore, the lack of consideration of the effects of complex microstructural characteristics significantly limits the accuracy of Ms prediction through modeling. Similar to the problem of Ms prediction, how to fully consider complex microstructure factors is a common problem in various prediction issues in the field of materials science; e.g., for prediction of strength in stainless steels, precipitation information should be considered [34], and for prediction of mechanical properties in RAFM steels(Reduced Activation Ferritic/Martensitic steels), phase information should be considered [35]. However, few studies reported methods for analyzing microstructure images with numerical data in the field of materials science. Therefore, developing a data-analysis method that can consider both microstructure images and numerical data is critically important for the development of materials science.

In this study, a deep learning (DL) strategy [36] was used to directly build the quantitative relationship between composition and microstructure images and Ms. Using the automatic image-information-extraction ability of a convolutional neural network (CNN) [37], complex microstructural characteristics can be directly used as input features for Ms prediction without the need to artificially convert images to numerical data. This is a critical improvement of the modeling methods in the field of materials science, as it cannot be achieved through traditional physical modeling methods. Therefore, this DL strategy, which considers the effects of complex microstructural characteristics, can provide significantly more accurate Ms prediction than can traditional models.

## 2. Modeling Process

### 2.1. Dataset Establishment

In this study, a medium-Mn steel data set was established. The compositions of the samples in the data set are listed in Table 1. An Mn content of 3–6 wt.% was used, and the contents of the other elements were maintained to be constant. The alloys were then smelted in a vacuum-induction furnace. The elemental contents were carefully tested using an infrared carbon–sulfur analyzer, a spectrophotometer, and an inductively coupled plasma emission spectrometer. Each ingot obtained after smelting was homogenized at 1200 °C for 5 h, then forged to a size of 120 mm × 150 mm. After forging, the alloys were subjected to hot rolling seven times and finally water-quenched to room temperature. Dilatometry specimens (10 mm × 4 mm × 2 mm) were prepared, and dilatometry experiments were performed with a DIL805A/D thermal dilatometer (TA Instruments, Hüllhorst, Germany). The final heat-treatment process is shown in Figure 1a, and the detailed parameters thereof are listed in Table 2. After normalization at 900 °C for 600 s, annealing was performed at different temperatures (735–790 °C) and for different times (0.5–20 min) depending on composition.

Figure 1b–e show representative microstructure images of the alloys with different compositions. All images obtained in this study were 1280 × 1280-pixel-value matrices with three channels. Each sample was related to one microstructure image, which means that the data set contained 24 microstructure images. Subsequently, the various Ms temperatures were obtained from the thermal expansion–temperature curve obtained from the dilatometry experiments. Figure 2 shows the effects of the annealing times on the Ms values for alloys with different compositions. Finally, a data set that comprised systematic compositions, microstructure images, and Ms information was established. Detailed information on this data set is provided in the Appendix A.

### 2.2. CNN Model

In this study, a CNN, many of which are widely used for classification and regression of images in the field of artificial intelligence (AI), was used for Ms prediction based on both composition and microstructure-image input. The framework used in this study is illustrated in Figure 3. In order to reflect the correlation between the composition data and the microstructure images, the contents of different main elements, namely C, Mn, and Si, were directly multiplied with the pixel value matrices of the related microstructure images. Then, the value matrices were cut into 224 × 224 submatrices to augment the amount of data. In addition, to meet the large data requirement of DL, further data augmentation was performed via 180° rotation or mirror transfer of the submatrices. Finally, 20,475 submatrices with integrated composition and microstructure information were obtained for the training, testing, and validation of the CNN model. In summary, the input data for the proposed CNN model is both composition (content of C, Mn, and Si) and microstructure images. The physical meaning of the input images is the microstructure factors, such as size and morphology of austenite, that have effects on the Ms. Figure 4 clearly shows the effect of each chemical composition: C, Mn, and Si. The absolute values of the Pearson correlations of all three elements were higher than 0.7, which means that the content of all three of these elements has a significantly high correlation with the Ms values. This analysis is also consistent with both previous experimental and computational results [15,23]. Therefore, it is reasonable to use these three elements as part of the main input in this research. After structure and parameter optimization, five convolutional layers, with a 3 × 3 convolution kernel, and five pooling layers were used for the CNN model. The fully connected layer had 512 neurons. Adam was chosen as the optimizer, and the learning rate was set to 3 × 10^−4^. For all of the aforementioned modeling of the CNN model and other AI models, data preprocessing and model training were implemented using Keras (version_2.4.3) and Scikit-learn (version_1.1.1) in Python (version_3.8.8). PyCharm (version_2021.1.3, generated by JetBrains from the Czechia) was used as the programming software in this research.

For the training and testing processes, 20% of the submatrices (4095 samples) in the data set were randomly selected as the validation set. The remaining submatrices were randomly divided into two parts 10 times, with 80% (13,104 samples) and 20% (3276 samples) of the submatrices used as training and testing sets, respectively. Finally, the predicted Ms output was obtained via passing the data through the convolutional, pooling, and fully connected layers. After these Ms predictions were obtained, the coefficient of determination (R^2^) and mean absolute error (MAE) were used to evaluate the performances of different models.

## 3. Results

### 3.1. Performance Optimization Results of CNN

To achieve the best performance of the CNN model, systematic optimization of the model structure and parameters is required. Figure 5 shows performance optimization results. First, the number of convolutional and pooling layers was optimized. As shown in Figure 5a, when the number of convolutional and pooling layers was only one, the average loss for both the training and testing set was relatively high because the structure was too simple to express the complex relationship between the composition/microstructure and the Ms. The loss in the model gradually decreased with increases in the number of layers. Finally, when five layers were used, the loss in the model became stable, which means that five convolutional and pooling layers could be the optimal structure of this model; fewer layers could reduce performance, and more layers could increase the risk of overfitting. After determination of the layer structure, three different optimizers were tested: namely, Adadelta, Nadam, and Adam. Figure 5b shows that Adam had better performance, with a lower average loss, for both the training and testing set. Therefore, the Adam optimizer was used in this study. Similarly, the effect of the learning rate was systematically analyzed, and the results thereof are shown in Figure 5c. The average loss became almost stable when a learning rate of 3 × 10^−4^ was used for the training set. In addition, at this learning rate, the performance of the testing set was the best. Therefore, the learning rate was set to 3 × 10^−4^ in this study. Finally, the stability and performance of this model were comprehensively evaluated during the iterative parameter optimization process. As shown in Figure 5d, the loss for both the training and testing set gradually decreased with an increase in the number of epochs. When the epoch number exceeded 200, loss stabilized below five, which confirmed the good convergence and stability of this model.

### 3.2. Prediction Results

The Ms prediction results from the CNN after parameter optimization are shown in Figure 6. These results demonstrate the excellent performance of the CNN model. The R^2^ values for the training, testing, and validation sets were all >0.99, which is much higher than those of most prediction studies that use AI methods in the field of materials science. Moreover, the R^2^ error bar for all of the sets was extremely small, indicating that this CNN model showed little risk of overfitting. The MAE results showed that the largest mean MAE for all the sets was only ~2 °C, which is lower than those in previous studies that also used DL strategies combined with complex deep data mining for Ms prediction. The CNN model developed in this study is much simpler than those of previous studies in terms of structural optimization and data preprocessing. In addition, the results for the testing and validation sets were nearly identical, which partly proves the extensibility of this model. Figure 6b–d show the detailed results of the Ms prediction. It is evident that the optimal prediction results for the samples in the training, testing, and validation sets fall precisely on a straight line with a slope of 1, which further confirms the stability and generalizability of this CNN model.

## 4. Discussion

### 4.1. Comparison with Traditional AI Methods That Only Use Composition Input

To explain the necessity of adding microstructure images as input, different traditional AI methods, including support vector regression (SVR), XGboost (XGB), random forest (RF), gradient boosting regression (GBR), and AdaBoost (Adb), were also used to predict Ms, with only composition and processing parameters as input. For the traditional AI methods, the data set with 32 samples was divided into training and testing sets at a ratio of 8:2. Figure 7 shows the results of the traditional AI methods. For all of the traditional AI methods, without the guidance of microstructure images, the mean MAE of the testing sets was much higher than that of the training sets, and the error bar of the MAEs for the testing sets was relatively large (Figure 7a). These results clearly indicate that the traditional AI methods showed a strong overfitting tendency. A similar conclusion can also be drawn based on the R^2^ results. The mean R^2^ values of the testing sets were lower, with larger error bars, than those of the training sets for all of the traditional AI methods, as shown in Figure 7b. Among the traditional AI methods, XGboost showed the lowest overfitting tendency (lower MAEs for both the training and testing set). The performance of XGboost regarding the training and testing sets is further shown in Figure 7c,d. Even for XGboost, which showed the best performance among the traditional AI methods, the mean MAE (3.02 °C) was higher than that of the CNN model built in this study (mean MAE = 2.11 °C), and the mean R^2^ of XGboost (0.982) was also lower than that of the CNN model (0.993).

In summary, compared with the traditional AI methods, the CNN model trained with microstructure images showed advantages in terms of both prediction accuracy and stability. With our understanding of materials science taken into account, it is reasonable that the CNN model had better performance because it contained more critical factors related to microstructure. In addition, in the field of AI, the better performance of the CNN model in this study could also be explained as follows: In this study, the data set contained only 24 samples with different compositions or processing details; therefore, the data set belonged to the class of extremely small-sample problems. Therefore, it is understandable that even SVR or XGboost, which are typically good at handling small-sample problems, could not completely solve the problem of overfitting in this study. However, with the addition of microstructure subimages, the 24 samples were augmented to 20,475 samples with more information. Therefore, this data augmentation greatly relieved overfitting and helped fully utilize the performance advantages of the DL. It should also be noted that the performance of SVR or XGboost can be improved through addition of more samples into the database. However, more samples mean more fabrication and sample preparation. Usually, traditional AI methods, such as SVR or XGboost, need at least hundreds of samples. This will lead to extremely high time and funding costs. Therefore, instead of addition of samples to the database, the CNN model, which only needs 24 samples, can be used as a more efficient way. In addition, compared with other traditional models, the main advantages of this CNN model are the efficiency of not only the small sample requirement but also the way that it can easily add complex microstructure factors into the model.

### 4.2. Comparison with Traditional CNN Model without Composition Input

As mentioned in Section 2, the content values of the different main elements (C, Mn, and Si) were directly multiplied with the pixel value matrices of the related microstructure images before being used as input in the CNN models. This differs from traditional CNN models, which directly use images as input. Therefore, to prove the necessity of using the data preprocessing step, a traditional CNN model with only microstructure subimages as input was also built for Ms prediction. The results for the traditional CNN models are shown in Figure 8. Although the mean R^2^ values of the testing and validation sets were slightly lower than that of the training set, both the gap and the error bars were small (Figure 8a). This indicates that with sufficient microstructure subimage data, the traditional CNN models could also overcome the problem of overfitting. Figure 8b–d show the prediction results of the traditional CNN models for the training, testing, and validation sets, respectively. It is clear that without use of composition information as input, the prediction accuracy of the CNN model was significantly degraded. For both the testing and validation set, many prediction results deviated significantly from a straight line with a slope of 1. It is widely accepted that Ms is determined through both composition and microstructural factors. Therefore, it is reasonable that consideration of only microstructural factors is not sufficient for accurate Ms prediction. In other words, the use of a data preprocessing step to combine both composition and microstructure information as input in the CNN model is necessary for Ms prediction.

### 4.3. Optimization of the CNN Framework for Addition of Composition Information

As mentioned above, both composition and microstructure information should be used as input to make accurate Ms predictions. However, instead of the data preprocessing step used in this study, composition information could also be added from the fully connected layer of the CNN model. Therefore, to build the optimal CNN framework for addition of the composition information, a CNN model in which composition information was added from the fully connected layer was also built for comparison. For this model, only the framework of the fully connected layer was changed, as shown in Figure 9. The composition vector was directly linked to the fully connected layer. Therefore, the composition layer could only affect the training of the parameters between the fully connected layer and the final output. The parameters in the convolutional and pooling layers, which determine feature extraction of images, were not affected by the composition in this framework. Figure 10a shows the optimal results from 10 random partitions of the data set. These results show that direct addition of composition information into the fully connected layer improved the performance of this CNN model (best MAE = 2.28 °C) over that of a traditional CNN model that did not consider the composition effect (best MAE = 3.48 °C). However, this performance improvement was less than that of the CNN model that used the data preprocessing step to add composition information (best MAE = 1.77 °C). The mean MAE and R^2^ values for all of the modeling methods are summarized in Figure 10b, which clearly shows the comprehensive advantages of the modeling strategy used in this study. This comparison’s results clearly show that the CNN model established in this research achieved a better performance because of the reasonable data preprocessing step and the reasonable architecture of the network for feature extraction. The data preprocessing step can help to fully reflect the interaction between composition (numerical data) and microstructure information (image data). Then, the CNN strategy could enable automatic extraction of the key features from the value matrix with both composition and microstructure information, leading to obtaining of accurate predictions of the Ms.

As mentioned in the introduction, various previous research has already established models that can clearly reflect the effect of composition on Ms. Therefore, the main advantage of this proposed CNN model is that it can easily add both composition and microstructure factors into the model, and it is a meaningful method with good development potential. However, as a newly proposed model, the expandability of this CNN model should be further evaluated in other issues with different databases. In addition, it should be mentioned that, similarly to most deep-learning models that use images as input, brightness, contrast, and signal-to-noise ratio will also probably affect the performance of this model. Therefore, in this research, the image database was carefully built with exactly the same SEM equipment, with constant testing parameters, to control this effect to an acceptable level. In order to further improve the robustness of this model, more modifications probably should be made. For instance, the contrast and brightness of the training images could be randomly adjusted in the acceptable range to enhance the robustness of the image quality [38,39,40,41]. In addition, EBSD(Electron Backscatter Diffraction) analysis could be employed to construct the high-quality data set in order to improve the accuracy of the feature extraction of the trained DL model for low-quality images with noise [42]. In addition, it was mentioned that the framework used in this research is available for small-sample databases (only 24 SEM images are needed). Therefore, it is reasonable to infer that this method can be transferred to other databases that need joint analysis of image and numerical data. Although further verification still should be made to clearly prove the extensibility of this model, it brings us a way worth trying for solving various small-sample problems in the field of materials science. In addition, in this special system of medium-Mn steels, with the combination of microstructure images and composition as input, this model can obtain better accuracy because it can consider the effect of morphology from images. Finally, it should be further clarified that, as an AI method, the CNN model established in this research is also a ‘black box’ strategy. This means that the advantage of strong data-analysis capability is at the expense of the physical meaning of the model. Therefore, all of the results shown in this research are meant to prove that this model can be used as a good Ms predictor but cannot be used to deepen the physical meaning of martensite transformation. This is also a common problem for nearly all of the AI models and needs to be solved in future work.

## 5. Conclusions

In order to fully consider the effect of microstructure on Ms, a CNN model with a specific data preprocessing approach was established for Ms prediction, and both composition and microstructure image information were used as input. The main conclusions thereof are as follows:(1)This CNN model made accurate predictions of the Ms values for medium-Mn steels. The MAE and R^2^ values of the validation sets were <2 °C and >0.99, respectively. This overcame the limitation that microstructures could not be digitized into numerical data or considered as factors in most previous models.(2)This CNN model offers significantly better prediction accuracy and stability than do traditional AI methods, especially decreasing the risk of overfitting, because the data preprocessing step used in this study enables data augmentation through use of microstructure images.(3)When a DL strategy is used to deal with small-sample problems for different data types, such as Ms prediction, using data preprocessing to obtain the value matrix that contains the interaction information of both numerical and image data is probably a better approach than directly linking the numerical data vector to the fully connected layer.(4)Although this CNN model is a powerful method for adding complex microstructure factors, its expandability should be further evaluated in other issues with different databases.

## Figures and Tables

**Figure 1 materials-16-00932-f001:**
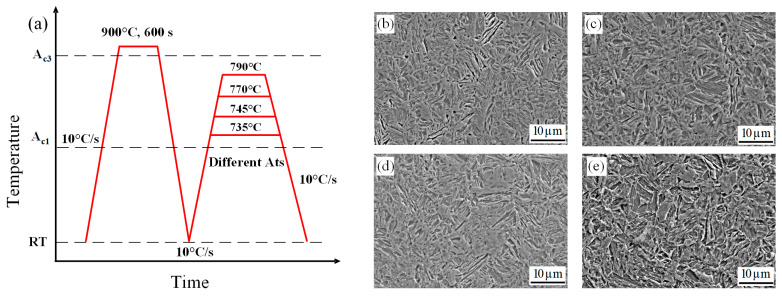
Final heat-treatment process and representative microstructure images of alloys with different compositions: (**a**) heat-treatment process, (**b**) Steel A, (**c**) Steel B, (**d**) Steel C, and (**e**) Steel D.

**Figure 2 materials-16-00932-f002:**
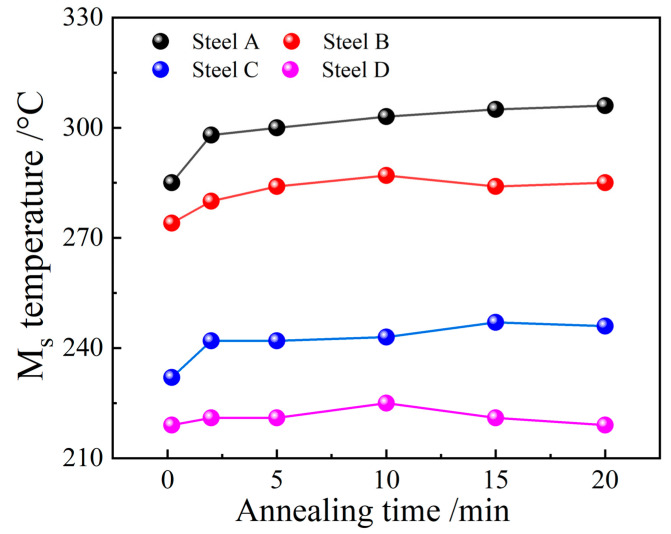
Effect of annealing time on the Ms values for alloys with different compositions.

**Figure 3 materials-16-00932-f003:**
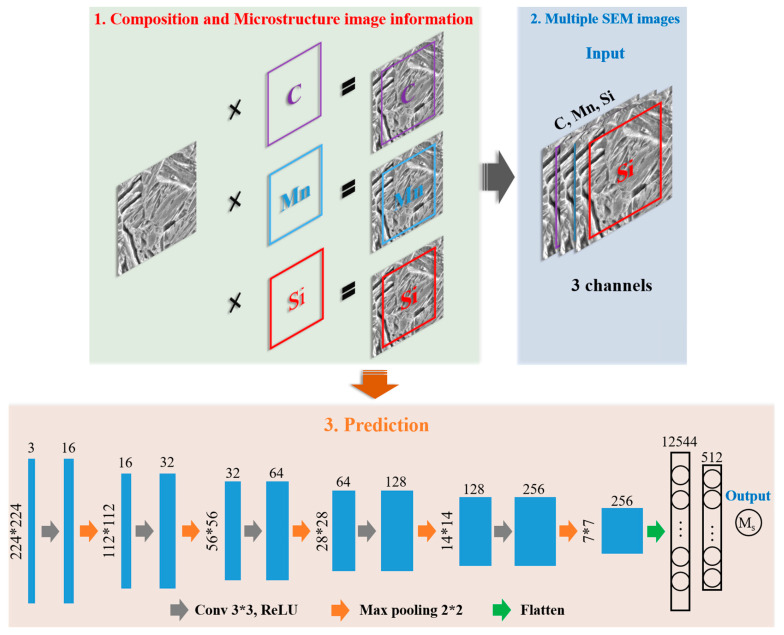
Framework of the CNN model for Ms prediction.

**Figure 4 materials-16-00932-f004:**
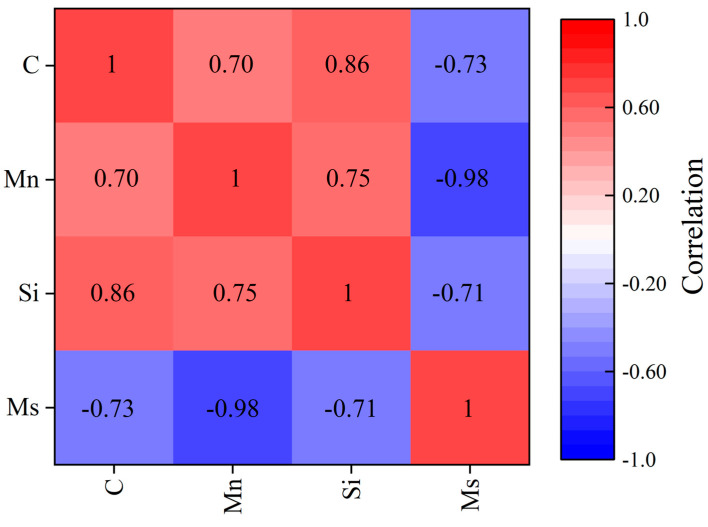
The Pearson-correlation-analysis results for composition.

**Figure 5 materials-16-00932-f005:**
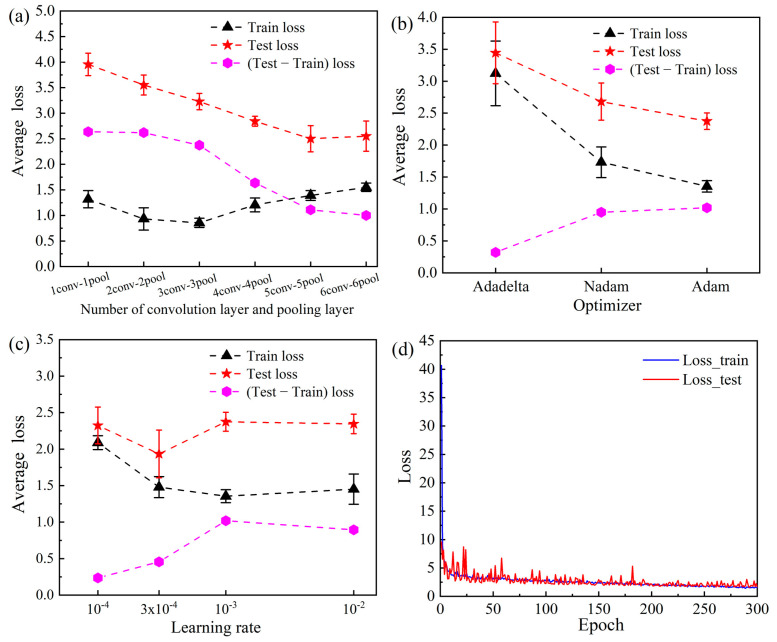
Performance optimization results: (**a**) number of convolution and pooling layers, (**b**) optimizer, (**c**) learning rate, and (**d**) epoch number.

**Figure 6 materials-16-00932-f006:**
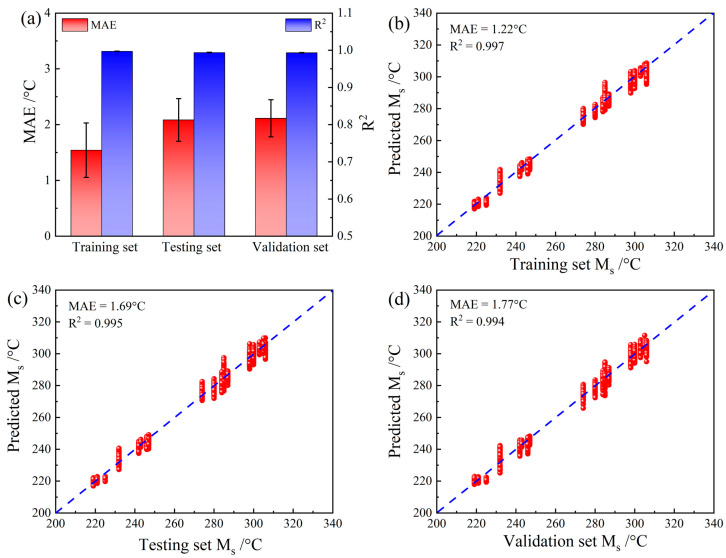
Results of Ms prediction using the CNN: (**a**) comparison of MAEs and R^2^ values, (**b**) results of the training set, (**c**) results of the testing set, and (**d**) results of the validation set.

**Figure 7 materials-16-00932-f007:**
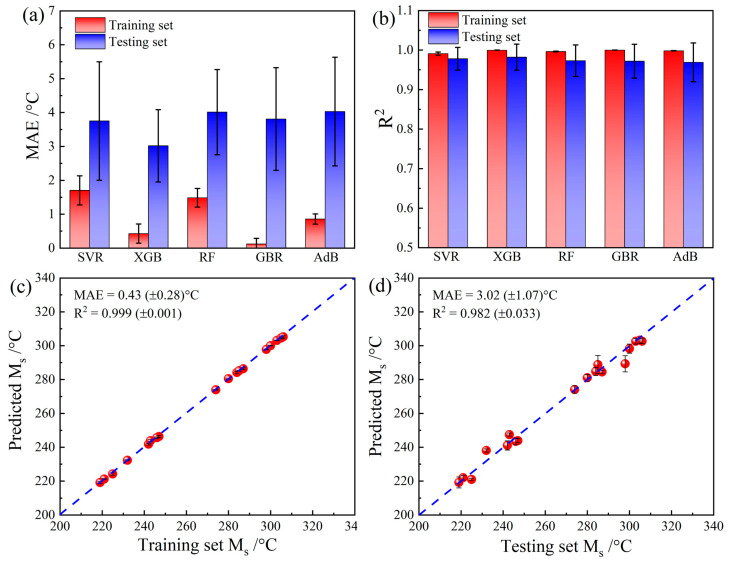
Comparison with traditional AI methods using only composition input: (**a**) comparison of MAEs, (**b**) comparison of R^2^ values, (**c**) results of the training set for XGboost, and (**d**) results of the testing set for XGboost.

**Figure 8 materials-16-00932-f008:**
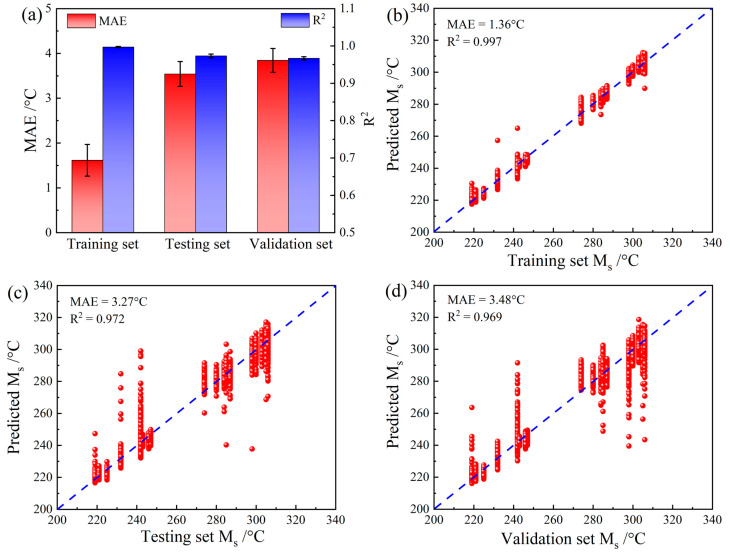
Comparison with traditional CNN methods using only microstructure image input: (**a**) comparison of MAEs and R^2^ values, (**b**) results of the training set, (**c**) results of the testing set, and (**d**) results of the validation set.

**Figure 9 materials-16-00932-f009:**
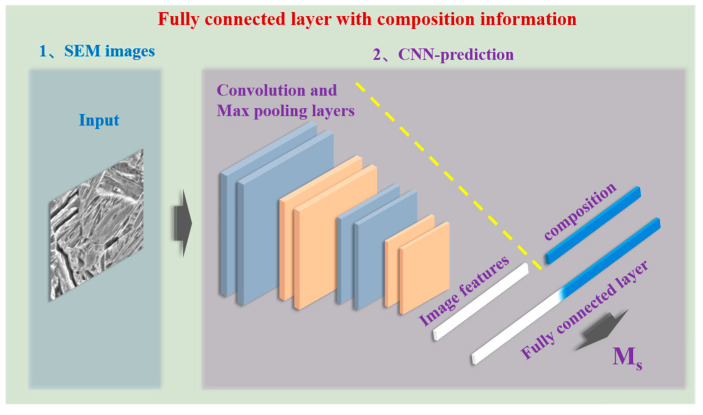
Framework of the CNN model after addition of microstructure information from the fully connected layer.

**Figure 10 materials-16-00932-f010:**
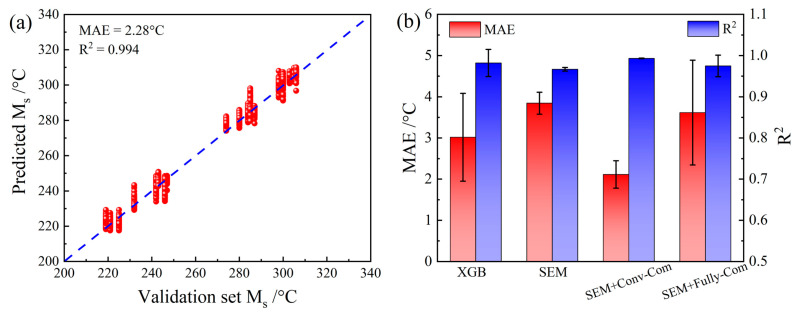
Results of the CNN model after addition of microstructure information from the fully connected layer: (**a**) results of the validation set and (**b**) comparison of MAEs and R^2^ values.

**Table 1 materials-16-00932-t001:** Composition of the samples in the data set (wt.%).

Parameter	Fe	C	Mn	Si
Steel A	Bal.	0.203	2.96	1.61
Steel B	Bal.	0.214	3.86	1.64
Steel C	Bal.	0.242	4.79	1.65
Steel D	Bal.	0.223	5.66	1.64

**Table 2 materials-16-00932-t002:** Detailed heat-process parameters at different annealing temperatures (ATs) and annealing times (Ats).

	A		B		C		D	
	AT/°C	At /min	AT/°C	At /min	AT/°C	At /min	AT/°C	At /min
Detailed Parameters	790	0.5	770	0.5	745	0.5	735	0.5
	790	2	770	2	745	2	735	2
	790	5	770	5	745	5	735	5
	790	10	770	10	745	10	735	10
	790	15	770	15	745	15	735	15
	790	20	770	20	745	20	735	20

## Data Availability

The data presented in this study are available upon request from the corresponding authors.

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
