# Peer review of "Martensite Start Temperature Prediction through a Deep Learning Strategy Using Both Microstructure Images and Composition Data"

_materials, 2023, doi:10.3390/ma16030932_

Round 1

Reviewer 1 Report

Abstract and Conclusion as to be reframed in such a way that both should be inline with each other

Introduction part can be improved by adding some more literatures, now it is difficult to get the actual novality of the work.

Actual research gap and Novality of the work to be clearly shown at the end of introduction part

All the Figures picture quality needs to be improved

Manuscripts to be thoroughly checked for grammatical mistakes.

Reviewer 2 Report

The article shows readers a potential application direction of deep learning in materials science, more specifically, the martensite start temperature (Ms) of steel. The authors have built a model with high accuracy because of using a deep learning strategy to deal with data containing both numerical and image data types, thereby approaching and handling the problem better.

The introduction is firstly the definition of the martensite start temperature (Ms) and the Linear empirical formulas studied in previous works, then, we can know that the internal mechanism of martensite transformation is much more complex than a linear relationship. However, during thermodynamic modeling, alloy systems can only be less than 10 elements due to limited error accumulation in multicomponent alloy systems. Using Machine learning can include many elements in a system, but ML has not solved the problem of the grain size and austenite morphology in steel, which is an important factor affecting the martensite start temperature (Ms). Therefore, the evaluation of a deep learning model can identify the issues. In particular, the authors leverage the autonomous image information extraction capability of a convolutional neural network (CNN) to demonstrate that complicated microstructural properties can be used directly as the input features for Ms prediction without the need to transform photos into numerical data. Although this introduction is quite reasonable, there are some errors in wording, for example, at the end of this opening paragraph, there are too many sentences using "however".

In the following section, modeling process, the authors describe the dataset establishment and definition of the CNN model to clarify the computational methods in this work.

The results section then states the important data obtained from the calculations of the CNN model, from which it is possible to predict Ms and  evaluate the performances of different models with MAE with R2.

In the next section, the discussion explains more about the obtained results, comparing the CNN model with the traditional AI method. Furthermore, it is a comparison of the improved CNN model with the traditional CNN model with a combination of both the composition and microstructure information. In this section, the authors compare SVR and XGboost to previous models such as XGboost, demonstrating that even SVR and XGboost, which are normally good at handling small-sample problems, cannot completely eliminate the overfitting problem. In addition, the CNN model trained with microstructure images demonstrated superior stability and prediction accuracy. So, why is it not feasible to increase the number of samples in the dataset to achieve better results (in comparison to the XGboost model)?

In general, the paper investigated an attractive topic about the application of deep learning to materials science and metallurgical engineering. The data and explanations are provided and organized rationally and logically; nonetheless, the authors should examine and correct a few small expression problems. In addition, the authors were able to enhance the models further and construct other data processing models for materials science and metallurgical engineering.

Other comments:

The statement "DiL805AD thermal dilatometer was used for testing Ms temperature" should be in details, or reference needed.

The temperature of heat treatment should be added in Figure 1.

"Detailed information on the dataset is provided in the supplementary material." should be provided? I cannot see it in the manuscript.

TRIP abbreviation should be explained before use.

How is the code written for CNN, Python or another language?

The references should follow the requirement from Materials. 

Figures with higher resolution should be updated.

Reviewer 3 Report

1.      In this theoretically research work author have nicely perform the equation anf this research work can give more accurate prediction. The model which used for prediction of Ms very carefully discuss in the systems and have very care ully simulate in the paper.

Reviewer 4 Report

I believe this manuscript should be accepted after minor revision.

Martensite start temperature (Ms) is predicted from the composition and microstructural images of Mn-steels using a deep learning (DL) method with a convolutional neural network model.  DL is an emerging method and, in addition to this, the authors have applied another new technique, in which the content value of each element is multiplied with the pixel values of microstructural images.  This contributes to the improvement in the prediction accuracy. 

The results are beneficial to both Ms prediction in materials designing and to developing new DL techniques.

I found only one English error, “was were ...” on 11th line in Section 2.2.

Reviewer 5 Report

1. In abstract the author must provide the highlight of this study

2. What are the advantages of employed technique?

3. What software or coding program was used to solve the numerical equations?

4. The authors should try to give advantageous of using of their method compared to others.

5. The authors need to explain that the numerical approach used in the research is one of the appropriate solutions in the context of the research problem. What are the achievements of previous studies based on a numerical basis? Also, describe what has not been achieved?

6. English language should be revised through the paper

7. The literature survey must be enhanced. There are still many papers in the allied fields that are not cited in introduction section.

8. There are many research papers study the same problem which investigated in the present paper. What is exactly the new point of this work?

9. Authors should explain more about the novelty of their work in introduction.

10. the solution method is not clear and needs to be clarified in depth in the contribution

11. In 'Results and Discussion' It is suggested that to provide physical explanations of all obtained results which can enrich the quality of the paper.

12. The conclusions should be clear and re-written; it must be enriched about discussion on solving problem.

Round 2

Reviewer 2 Report

This manuscript's revised version is significantly improved. I suggested that this one be published.

Reviewer 5 Report

Authors have made all modifications